# Evaluation of Lysophosphatidic Acid Effects and Its Receptors During Bovine Embryo Development

**DOI:** 10.3390/ijms26062596

**Published:** 2025-03-13

**Authors:** Bo Yu, Shuying Dai, Lei Cheng, Qirong Lu, Qing Liu, Hongbo Chen

**Affiliations:** 1Laboratory of Genetic Breeding, Reproduction and Precision Livestock Farming, School of Animal Science and Nutritional Engineering, Wuhan Polytechnic University, Wuhan 430023, China; wonderfish@whpu.edu.cn (B.Y.); dsy_0109@163.com (S.D.); liu3996406@163.com (Q.L.); 2Hubei Provincial Center of Technology Innovation for Domestic Animal Breeding, Wuhan Polytechnic University, Wuhan 430023, China; 3Institute of Animal Science and Veterinary Medicine, Wuhan Academy of Agricultural Sciences, Wuhan 430208, China; chenglei@wuhanagri.com; 4Wuhan Engineering and Technology Research Center of Animal Disease-Resistant Nutrition, School of Animal Science and Nutritional Engineering, Wuhan Polytechnic University, Wuhan 430023, China; qirongluvet@whpu.edu.cn

**Keywords:** lysophosphatidic acid, bovine, in vitro embryo production, receptors, apoptosis

## Abstract

Lysophosphatidic acid (LPA) is a small bioactive phospholipid which plays an important role during embryonic development and promotes developmental potential of in-vitro-produced (IVP) embryos in several species, including sheep and pigs. In bovines, LPA accelerates IVP blastocyst formation through the Hippo/YAP pathway. However, other LPA effects and its potential receptors during bovine embryo development are less clear. In this study, we used enzyme-linked immunosorbent assay (ELISA) to assess the presence of LPA in bovine oviductal fluid and determine cell apoptosis in embryos after LPA stimulation by terminal deoxynucleotidyl transferase dUTP nick end labeling (TUNEL) assay and quantitative reverse transcription polymerase chain reaction (qRT-PCR). We further evaluated potential receptors of LPA through molecular docking, RNA-seq data analysis and quantitative RT-PCR. LPA was found to be present in oviductal fluid. An increase in total cell number and a decrease in apoptosis levels were detected in day 7 blastocysts after LPA treatment. Among eight LPA receptors (LPARs), GPR87 and LPAR2 showed the highest affinity with LPA and their transcripts were expressed in embryos after the 16-cell stage in RNA-seq and qRT-PCR analysis. However, only the expression of *LPAR2* was significantly increased in day 6 blastocysts after LPA stimulation, indicating its potential role in LPA-mediated signaling pathways. Our data highlight the positive effects of LPA on embryos and enrich information of related signaling mediators of LPA during embryonic development.

## 1. Introduction

Lysophosphatidic acid (LPA) is a simple bioactive glycerophospholipid that regulates diverse cellular effects on different cell types [1,2]. LPA is predominantly produced by two enzymes called autotaxin (ATX) and phospholipase A (PLA) [3,4]. In the past few decades, accumulating evidence has highlighted the important roles of LPA in vertebrate reproduction [5,6]. It has been reported that LPA is involved in the regulation of sperm motility through glycolysis enhancement and L-type calcium channels activation in humans [7]. In rats, LPA has been found to stimulate uterine contraction in vitro and its effect does not lead to the de novo synthesis of prostaglandin [8].

In vitro embryo production (IVP) is currently one of the most important assisted reproductive technologies and has been widely used to improve production efficiency of superior genotypes [9,10]. Although the blastocyst rate has been increased over the years, the developmental potential of IVP embryos is still much lower compared to their in vivo counterparts [11]. Recent investigations have demonstrated that LPA plays an important role during embryonic development and promotes developmental potential of IVP embryos in several aspects [6,12]. Studies in sheep have shown that LPA supplementation enhances the maturation and blastocyst rate, promoting the expression of *CDX2* and *OCT4* in IVP embryos [13]. Moreover, in porcines, LPA can improve oocyte maturation and developmental potential of IVP embryos, resulting in apoptosis reduction in either IVP or parthenogenetic activation embryos [14]. In addition, our previous data suggest that LPA induced bovine IVP blastocyst formation in advance and enhanced YAP expression [15]. However, there is a lack of information about the molecular mechanisms of LPA on early IVP embryos.

It has been well documented that LPA mediates diverse biological actions through binding to and activating G-protein-coupled receptors, namely LPA receptors (LPARs) [1,16,17]. Based on their homology, LPAR1, LPAR2 and LPAR3 belong to the endothelial differentiation gene (EDG) family, which share 50–57% identities at the amino acid level; while LPAR4, LPAR5 and LPAR6, are non-EDG receptors with 35–55% amino acid identity [18,19]. GPR87 and P2Y10 are two newly identified LPA receptors, which both display tissue specific expression [1]. It has been reported that deletion of LPAR3 in mice leads to delayed implantation and changes in embryo spacing and size, which indicates its essential role during embryo implantation [20].

Even though different LPAR subtypes have been identified, expression patterns of LPARs during embryonic development are not uniformly conserved among species. In sheep, *LPAR1–3* expressions were detected in trophectoderm, with *LPAR1* and *LPAR3* peaking on day 14 of pregnancy [21]. In cows, the mRNA abundance of *LPAR1–4* was higher in day 5 embryos compared with that in day 8 blastocysts [22]. During mouse blastocyst development, *Lpar1* was consistently expressed, while expression of *Lpar2* only existed in late-stage blastocysts and expression of *Lpar3* was absent [23].

Here, we took a multifaceted experimental approach to study the effects and receptors of LPA during bovine preimplantation development. We assessed the presence of LPA in oviductal fluid and checked apoptosis levels in embryos after LPA treatment. We further examined the potential receptors of LPA by molecular docking, quantitative reverse transcription PCR (qRT-PCR) and analysis of RNA-seq data. Our data provide basic information on the effects of LPA and the receptors involved in LPA-mediated signaling pathways.

## 2. Results

### 2.1. Presence of LPA in Oviductal Fluid

The oviductal fluid is the primary microenvironment for early embryos and nourishes embryos during preimplantation development. As our previous data indicated that LPA accelerates bovine IVP blastocyst formation [15], we presumed that LPA is a necessary functional molecule for bovine embryo development. To check the presence of LPA in microenvironmental conditions where in vivo bovine embryos develop, we quantified the concentration of LPA in oviductal fluid, follicular fluid, uterine fluid and serum using enzyme-linked immunosorbent assay (ELISA). As expected, LPA was present in all four fluids with concentrations of 8.65 μmol/L, 8.94 μmol/L, 8.99 μmol/L and 8.97 μmol/L in oviductal fluid, follicular fluid, uterine fluid and serum, respectively (Figure 1A).

We next evaluated the gene expression levels of LPA production enzymes in oviduct epithelial cells, granulosa cells and uterine epithelial cells. Expressions of *ATX* and *PLA2* were both detected in all three cell types (Figure 1B,C). Notably, expression of *ATX* was significantly higher in oviduct epithelial cells than in granulosa cells (Figure 1B), while higher expressions of *PLA2* were detected in uterine epithelial cells (Figure 1C) compared to granulosa cells. Combined, the data suggest that LPA is present in the environmental condition where in vivo embryos grow.

### 2.2. The Level of Apoptosis in Embryos After LPA Stimulation

As our previous data indicated that LPA accelerates bovine IVP blastocyst formation [15], we presumed that LPA stimulation enhances embryo quality, and the level of apoptosis is a standard marker of embryo quality. We performed terminal deoxynucleotidyl transferase dUTP nick end labeling (TUNEL) to identify apoptotic cells in bovine embryos after LPA treatment (Figure 2A). We first used DAPI to examine the total cell number from day 4 16-cell-stage embryos to day 7 blastocysts. Significantly higher total cell number was detected in day 7 blastocysts after LPA treatment, with an average total cell number of 140.1 and 178.1 in the control and LPA group, respectively (Figure 2B). We then evaluated the level of apoptosis in embryos by measuring TUNEL-positive cells and TUNEL intensity of embryo area. The percentage of TUNEL-positive cells was significantly lower in day 7 LPA exposure blastocysts (1.5%) compared to control medium (4.7%) (Figure 2C). Consistently, a significantly lower intensity value of TUNEL was detected in embryos cultured with LPA, compared to control medium for day 7 blastocysts (342465.1 vs. 818696.3) (Figure 2D). We further examined the expression of *BAX* (pro-apoptotic regulator) and *BCL2* (anti-apoptotic regulator) in embryos. The expression level of *BAX* was similar between embryos cultured with and without LPA from day 4 16-cell-stage embryos to day 7 blastocysts (Figure 2E). To our expectation, a significantly higher expression of *BCL2* was detected in day 7 blastocysts after LPA stimulation compared to the control group (Figure 2F). Combined, the data suggest that LPA stimulation improves day 7 blastocyst quality in respect of total cell number and apoptosis level.

### 2.3. Molecular Docking Between LPA and LPARs

At least eight LPARs (LPAR1–6, GPR87 and PRY10) have been identified in humans or mice, but their interactions with LPA have not been studied in bovines. To explore the potential ligand–receptor interactions between LPA and LPARs, we performed a molecular docking study. All eight LPARs exhibited overall favorable binding modes with total scores exceeding 6 (Table 1). Notably, GPR87 (total score 13.2981) had the highest binding affinity with LPA compared the other seven LPARs but its crash score reached −5.6552, indicating a relatively high chance of inappropriate penetration into LPA. The second most active LPAR is LPAR2 with a total score of 11.4654 and a crash score of 2.3193, followed by P2Y10 with a total score of 10.0559 and crash score at 2.2346. Overall, GPR87, LPAR2 and P2Y10 were identified as high-affinity receptors for LPA and the pose of the eight LPARs and the docking position of their targets with LPA are shown in Figure 3.

### 2.4. Expression of LPARs During In Vivo and In Vitro Preimplantation Embryo Development

To systematically assess gene expression of different LPARs during bovine embryogenesis, we analyzed RNA-seq data generated from in vivo (GSE59186) and in vitro (GSE52415) embryos at different stages [24,25]. Expressions of *LPAR1*, *LPAR2*, *LPAR3*, *LPAR6* and *GPR87* were detected in both in vivo and in vitro embryos. Expression of *LPAR1* was relatively constant throughout embryonic development but was elevated at the early morula stage in in vivo embryos, while its expression increased after the 16-cell stage and reached a high level at the blastocyst stage in in vitro embryos (Figure 4A,B). The expression of LPAR2 was relatively stable during in vitro embryo development, while its expression increased from GV oocyte to blastocyst in in vivo embryos. Expression of *LPAR3* decreased after the 4-cell stage both in in vivo and in vitro embryos. Expressions of *LPAR4* and *LPAR5* were below the detectable level throughout in vivo and in vitro embryo development. Expression of *LPAR6* was low before the 8-cell stage but was elevated at the 16-cell and the compact morula stages in in vivo embryos. In in vitro embryos, Expression of *LPAR6* had a trend of increase from the 8-cell stage to the 16-cell stage. Expression of *GPR87* increased after the 4-cell stage and reached the highest level at the 16-cell stage in both in vivo and in vitro embryos. Expression of *P2RY10* was below the detectable level in both in vivo and in vitro embryos.

### 2.5. Expression of LPARs in In Vitro Embryo After LPA Treatment

To evaluate gene expressions of different LPARs after LPA stimulation and validate RNA-seq findings, we next quantified gene expressions for different LPARs using qRT-PCR. Here, we focused on the period between the 16-cell stage and the blastocyst, since we found that embryo development and apoptosis level change after the 16-cell stage when cultured with LPA. Expressions of *LPAR1*, *LPAR2*, *LPAR3* and *GPR87* were detectable (Figure 5A–D), but *LPAR4* and *LPAR5* transcripts remained undetectable throughout the examined developmental stages, which were consistent with the RNA-seq results. Notably, we identified a discrepancy between two detection methods: while *LPAR6* expression was consistently observed in both in-vivo- and in-vitro-derived embryos in RNA-seq data, *P2RY10* mRNA was exclusively detectable as determined by qRT-PCR (Figure 5E). A particularly significant finding emerged in *LPAR2* expression analysis. The expression of *LPAR2* was significantly higher in day 6 LPA-stimulated blastocysts compared to control blastocysts and its expression increased significantly from day 5 morulae to day 6 blastocysts after LPA treatment (Figure 5B).

## 3. Discussion

LPA is a simple bioactive phospholipid, with multiple functions during early embryo development in domestic animals. We previously reported that LPA accelerates bovine in-vitro-produced blastocyst formation through the Hippo/YAP pathway [15]. However, the specific receptors mediating LPA’s effects remain uncharacterized in bovine embryos. Here, we demonstrate that LPAR2 potentially plays a predominant role for mediating LPA signaling involved in proliferation and apoptosis in bovine embryos.

The oviductal fluid is considered the most suitable microenvironment for early embryo development. In this study, LPA is present in oviductal fluid and the LPA-producing enzyme AXT gene is highly expressed in oviduct epithelial cells, indicating that LPA may be a necessary molecule for bovine embryo development. However, LPA is not supplied in standard protocols of bovine embryo culture, providing a clue that the low quality of IVP embryos might be related to the absence of LPA. Interestingly, we also detected LPA levels in follicular fluid and serum and LPA-producing enzyme gene expression in granulosa cells and uterine epithelial cells. It would be of physiological significance to study how LPA participates in oocyte maturation or estrus cycles by determining LPA levels in different follicle classes or based on different estrus stages.

Embryos with a faster development are usually associated with better quality and result in a higher pregnancy rate [26,27,28]. Our previous data have demonstrated that LPA accelerates the bovine developmental process and there is a higher blastocyst rate on day 5 and day 6 of in vitro culture [15]. Therefore, we hypothesized that LPA stimulation in bovine embryos would result in better embryo quality. Indeed, in this study, we show that culturing bovine embryos in the presence of LPA increases the total cell number and decreases the apoptosis level, which are indicators of good-quality embryos. It has been reported that LPA treatment increases the blastocyst rate in both sheep and porcine embryos [13,14]. In porcine embryos, consistent with our results, an increase in the total cell number and decrease in the apoptosis level were also detected after LPA exposure [14]. Since the high blastocyst rate, high embryonic cell count and low apoptosis level are hallmark indicators for good-quality embryos, we therefore suggest that LPA improves developmental competence of early embryos. For accurate assessment of the developmental potential of embryos after LPA treatment, future work could aim at verifying a higher pregnancy rate and the birth of live offspring after transferring LPA-treated embryos to recipient females.

Molecular docking is a computational technique that analyses the binding affinity of small molecules to receptor proteins, which is usually used as a formidable tool for drug development [29]. Since LPA is a small molecule with a molar mass of 436.52 g/mol, we used this algorithm to predict and rank the LPA receptors according to their binding affinity. The docking outcomes are evaluated by the scoring functions total, crash and polar score. Higher total score values indicate strong binding affinity and negative values suggest improper ligand positioning [30]. According to the total score, all LPA receptors show moderate to good binding scores in terms of affinity to LPA, with GPR87, LPAR2 and P2RY10 exhibiting the highest interaction potential (total score >10). However, compared to LPAR2 and P2Y10, GPR87 shows a substantially negative crash score, indicating its higher chance of inappropriate penetration into the binding site [31].

To sort out physiologically relevant receptors of LPA during bovine embryo development, normalized read count data generated from in vivo and in vitro embryos at different stages were studied [24,25]. We found that expression profile patterns of *LPAR1*, *LPAR2* and *LPAR6* were different between in vivo and in vitro embryos. Since LPA is absent in in vitro embryo culture conditions but is present in bovine oviductal fluid, we suggest different expression patterns of LPARs are caused by the different concentrations of LPA in the embryo microenvironment. Consistent with our molecular docking results, both *LPAR2* and *GPR87* were expressed in in vivo and in vitro preimplantation embryos. Expression of *LPAR2* and *GPR87* increased after the 16-cell stage and the 8-cell stage respectively. Coincidentally, our previous data show that bovine embryos stimulated with LPA change embryonic development only after the 16-cell stage [15], suggesting LPAR2 and GPR87 are potential receptors of LPA during embryo development in bovines.

In mouse embryos, *Lpar1* is constitutively expressed but the expression of *Lpar2* is restricted to late-stage blastocysts while expression of *Lpar3* is below detectable level [23]. *LPAR1* and *LPAR3* were expressed during early pregnancy and displayed a peak on day 14 in sheep [21]. In bovine, transcripts of LPAR1–4 are detectable and display higher expression in day 5 embryos compared to day 8 embryos [22]. Our study provides the first comprehensive analysis of all eight LPAR transcripts throughout bovine embryonic development. Expressions of *LPAR4*, *LPAR5* and *LPAR6* were below the detectable level throughout bovine embryo development, which are consistent with RNA-seq results except *LPAR6*. Primers of *LPAR4*, *LPAR5* and *LPAR6* were validated in somatic cells to avoid technical problems. However, we did not observe the downregulation of *LPAR1–3* from day 5 embryos to day 7 embryos, which might be due to the downregulation only occurring after day 7. Most notably, we uncovered that the expression of *LPAR2* was significantly higher in day 6 blastocysts after LPA treatment, indicating its potential role of mediating LPA signaling. To confirm the role of LPAR2, future work could focus on how the embryo development changes by using a specific inhibitor or siRNA for LPAR2. Moreover, we provided information on five binding sites between LPA and LPAR2 after molecular docking. It would be interesting to mutate one or more codons encoding the amino acid of the binding sites, providing us clues on how LPA interacts with LPAR2.

## 4. Materials and Methods

### 4.1. In Vitro Embryo Production and LPA Treatment

All reagents and chemicals were obtained from Sigma Aldrich (St. Louis, MO, USA), unless otherwise specified.

Cow ovaries obtained from slaughterhouses were immediately transported to the laboratories in a prewarmed thermos flask and washed in 0.9% NaCl solution containing penicillin/streptomycin (PS) (100 µg/mL) (Gibco, Waltham, MA, USA) at 30 °C. Follicular fluid was aspirated from antral follicles (2–8 mm diameter) using a sterile winged infusion needle (0.8 × 28 mm). The fluid was collected in sterile 50 mL centrifuge tubes. The sedimented components of the liquid were transferred to a 60 mm culture dish and the cumulus–oocyte complexes (COCs) with multilayered cumulus cells were identified under a stereomicroscope.

Selected COCs underwent in vitro maturation for 23 h at 38.8 °C in a humidified atmosphere of 6% CO_2_. Then, the matured oocytes were inseminated with 1 × 10^6^/mL frozen–thawed bovine spermatozoa for 18–22 h and the day of fertilization was considered as day 0 of embryogenesis. Following fertilization, presumptive zygotes were denuded by vortexing for 3 min. The zygotes were randomly allocated into two experimental groups: (i) control group, cultured in BO-IVC medium (IVF Bioscience, Cornwall, UK); and (ii) LPA group, cultured in BO-IVC medium with 10^−5^ M LPA. Embryos were cultured at 38.8 °C, in a humidified 6% CO_2_ and 6% O_2_ atmosphere.

### 4.2. Sample Collection

After rinsing in PBS three times, 8-cell embryos, morulae, early blastocysts and blastocysts were collected on day 4, day 5, day 6 and day 7 after the start of fertilization, respectively. Bovine oviduct, ovary, uterine and blood samples were collected from the slaughterhouse, followed by isolation of bovine oviductal fluid, follicular fluid, uterine fluid, serum, oviduct epithelial cells, granulosa cells and uterine epithelial cells.

### 4.3. TUNEL Assay

Apoptotic cells were detected using a TUNEL assay kit (Beyotime, Shanghai, China) following standardized protocols. In short, embryos were immersed in 4% paraformaldehyde for 30 min. After three PBS washes, embryos were treated with 0.3% Triton X-100 for 5 min at room temperature. After washing three times in PBS, samples were incubated in TUNEL assay solution at 37 °C for 60 min in the dark. Samples were sealed with anti-fade mounting medium with DAPI (Beyotime, Shanghai, China) after washing. Fluorescent images were obtained using a fluorescence microscope (Thermo Fisher Scientific, Waltham, MA, USA) with Z-stack acquisition and image quantification was conducted using ImageJ software (v1.56p) (National Institutes of Health, Bethesda, MD, USA).

### 4.4. LPA Concentration Detection

To determine the concentrations of LPA in tubal fluid, uterine fluid, follicular fluid and serum, a bovine LPA ELISA kit was used (Xingyu biotechnology, Shanghai, China). The samples were first centrifuged at 3000× *g* for 10 min to remove the pellet and polymer. The tin foil was removed from a pre-coated plate after equilibrating for 20 min at room temperature. To set up standard wells and sample wells, 50 μL of different concentrations of standards was added to each standard well, and 10 μL of the test solution mixed with 40 μL of sample diluent was added to the sample wells. Subsequently, 100 μL of horseradish-peroxidase-labeled detection antibody was added to wells and incubated at 37 °C for 60 min. Liquid in wells was discarded, followed by washing 5 times. The substrate was added per well and the plate was incubated at 37 °C in the dark for 15 min. The OD values of each well were measured at 450 nm, and the concentrations of LPA were calculated according to the standard curve of standard wells.

### 4.5. RNA Extraction, cDNA Synthesis and Quantitative Reverse Transcription PCR

Groups of 15–18 embryos were homogenized in 100 µL RLT buffer (Qiagen, Duesseldorf, German) and stored at −80 °C until RNA isolation. Total RNA extraction and genomic DNA elimination were performed using the RNeasy Micro Kit (Qiagen) according to the manufacturer’s instructions. After RNA isolation, reverse transcription was conducted using the PrimeScript™ RT reagent Kit (Takara, Tokyo, Japan). The reverse transcription (RT) mixture contained 10 µL of the RNA sample, 1 µL of PrimeScript RT Enzyme Mix I (Takara), 1 µL of RT Primer Mix (Takara), 4 µL of 5×PrimeScript Buffer 2 (Takara) and 4 µL of RNase-free dH_2_O (Takara), in a total volume of 20 µL. Gene-specific primer pairs (Sangon Biotech, Wuhan, China) were designed using Primer Premier (v5.0) and the NCBI primer-blast tool (Table 2). The qRT-PCR mixture contained 5 µL iQ TB Green (Takara), 3.6 µL DNase/RNase-free water (Takara) and 1 µL cDNA with a final primer concentration of 10μM. Reactions were performed in a QuantStudio™ 1 Plus System (v1.1) (Thermofisher, Waltham, MA, USA) with the primer-specific annealing temperature (Table 2) based on the manufacturer’s protocol. The PCRs were carried out on three biological replicates in duplicate.

### 4.6. Molecular Docking

Protein sequences of different LPARs were obtained using NCBI-BLAST (Basic Local Alignment Search Tool) (https://blast.ncbi.nlm.nih.gov/Blast.cgi, accessed on 11 March 2025). The crystal structures of the target proteins were subsequently simulated using the SWISS-MODEL. The target proteins were then refined using Sybyl-X software (v2.0) (Tripos, St. Louis, MO, USA), which involved the removal of water molecules and metal ions, the addition of hydrogen atoms, the correction of any missing amino acid residues within the crystal structure and other structural optimizations. LPA structure was obtained from the PubChem database. Evaluation of the direct binding capacity of LPARs to LPA was performed using Sybyl-X. Visualization of molecular docking results was carried out using the PYMOL website (https://www.pymol.org).

### 4.7. RNA-Seq Data Processing

Raw RNA-seq data were obtained from GEO Series GSE52415 (in vitro embryos) and GSE59186 (in vivo embryos). The quality of raw sequence data was checked using FastQC (v1.0.0), followed by low-quality read and adapter sequence removal using FastP (v0.24.0). The filtered clean reads were aligned to the Bos taurus reference genome ARS-UCD1.3 (GCA_002263795.3) using Hisat2 (v2.1.0). Subsequently, the resulting SAM files were read and converted to BAM files using Samtools (v1.21). The read counts matrix was calculated using featureCounts (v 2.0.5). Fragments per kilobase of transcript per million mapped reads (FPKM) values of genes were calculated in edgeR (v4.4.2).

### 4.8. Statistical Analysis

All data processing and calculations were conducted using Microsoft Excel (v16.0) and subsequent statistical analysis was performed using GraphPad Prism (v9.5). For pairwise comparisons between two groups, a two-tailed unpaired Student’s *t*-test was applied under assumption of equal variance. Comparisons between multiple groups were analyzed using a one-way analysis of variance (ANOVA) followed by Tukey’s honestly significant difference (HSD) post hoc test. Statistical significance threshold was set at *p* < 0.05.

## 5. Conclusions

This study has demonstrated that LPA is present in oviductal fluid and LPA stimulation increases blastocyst quality in terms of apoptosis level and total cell number. Among LPA receptors, GPR87, LPAR2 and P2Y10 displayed the top 3 highest affinities with LPA. Expression of *LPAR2* and *GPR87* increased after the 8-cell stage according to RNA-seq data. However, only *LPAR2* expression increased in day 6 blastocysts after LPA treatment, suggesting that LPAR2 is potentially the main receptor involved in LPA-mediated signaling pathways.

## Figures and Tables

**Figure 1 ijms-26-02596-f001:**
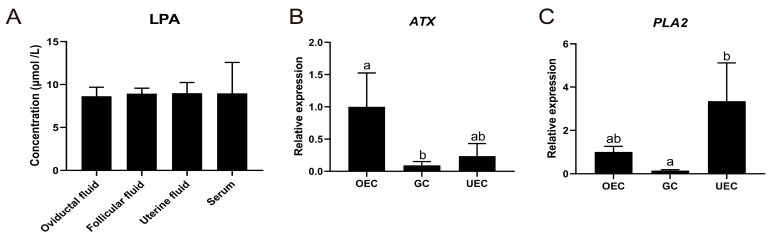
Lysophosphatidic acid (LPA) detection in oviductal fluid and LPA. (**A**) Concentration of LPA in oviductal fluid, follicular fluid, uterine fluid and serum. The relative expression of *ATX* (**B**) and *PLA2* (**C**). Error bars indicate standard deviations of three biological replicates. Significant differences between different samples are presented by different letters. OEC = oviduct epithelial cells, GC = granulosa cells, UEC = uterine epithelial cells.

**Figure 2 ijms-26-02596-f002:**
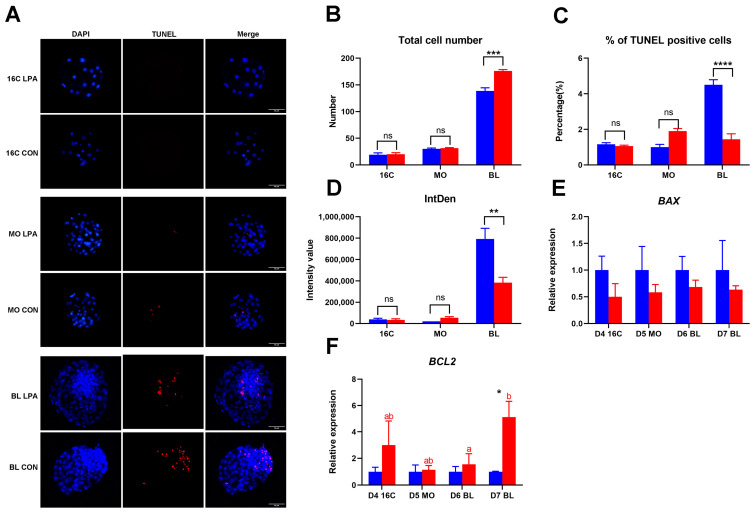
The level of apoptosis in embryos cultured in the absence of LPA (blue column) and in the presence of LPA (red column). (**A**) terminal deoxynucleotidyl transferase dUTP nick end labeling (TUNEL) staining of day 4 16-cell embryos, day 5 morulae and day 7 blastocysts cultured in the absence of LPA and in the presence of LPA, scale bar = 50 μm. Total cell number (**B**), percentage of TUNEL-positive cells (**C**), IntDen values of embryo area (**D**), the relative expression of *BAX* (**E**) and *BCL2* (**F**) in day 4 16-cell embryos, day 5 morulae and day 7 blastocysts cultured in the absence of LPA and in the presence of LPA. * (*p* < 0.05), ** (*p* < 0.01), *** (*p* < 0.001) and **** (*p* < 0.00001) indicate a significant difference between embryos cultured with and without LPA. Significant differences between developmental stages are presented by different letters with the same color (*p* < 0.05). Relative expression from control embryos was set at 1. Error bars indicate standard deviations of three biological replicates. D = day, 16C = 16-cell-stage embryo, MO = morula, BL = blastocyst.

**Figure 3 ijms-26-02596-f003:**
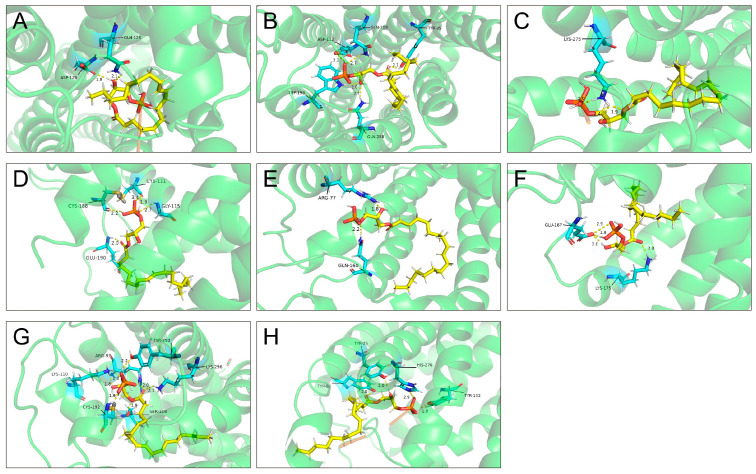
Molecular docking between LPA (green) and LPARs (yellow). (**A**) LPAR1, (**B**) LPAR2, (**C**) LPAR3, (**D**) LPAR4, (**E**) LPAR5, (**F**) LPAR6, (**G**) GPR87, (**H**) P2Y10. Binding amino acid residues are shown in blue.

**Figure 4 ijms-26-02596-f004:**
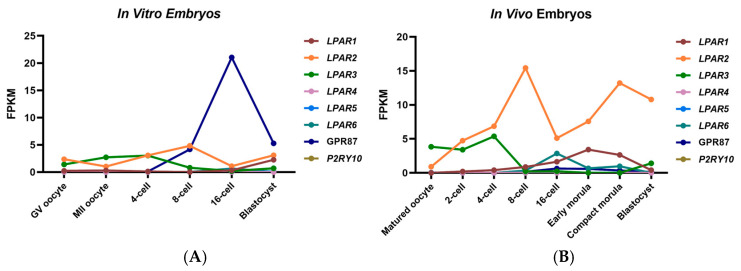
RNA levels of LPARs during in vitro (**A**) and in vivo (**B**) preimplantation embryo development, as determined by the reanalyzed RNA-seq data [24,25]. FPKM = fragments per kilobase of transcript per million mapped reads.

**Figure 5 ijms-26-02596-f005:**
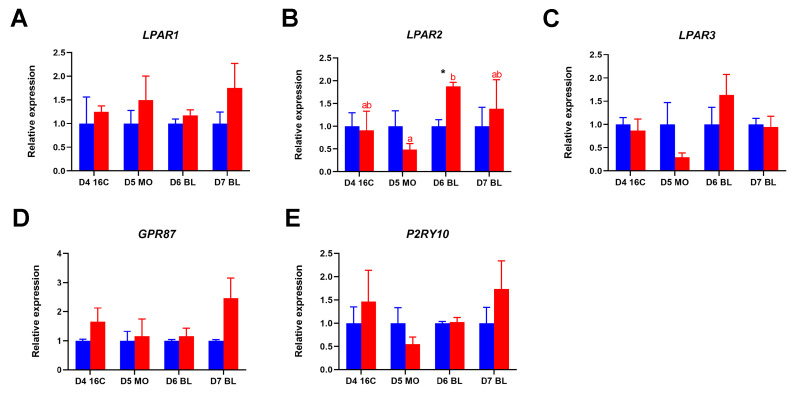
The relative expression of LPAR genes in bovine embryos cultured in the absence of LPA (blue column) and in the presence of LPA (red column), as determined by quantitative reverse transcription polymerase chain reactionquantitative (qRT-PCR). (**A**) *LPAR1*, (**B**) *LPAR2*, (**C**) *LPAR3*, (**D**) *GPR87*, (**E**) *P2RY10*. Relative expression from control groups was set at 1; * (*p* < 0.05) indicates a significant difference between embryos cultured with and without LPA. Significant differences between developmental stages are presented by different letters with the same color (*p* < 0.05). Error bars indicate standard deviations of three biological replicates. D = day, 16C = 16-cell stage embryo, MO = morula, BL = blastocyst.

**Table 1 ijms-26-02596-t001:** Binding scores (total score, crash score and polar score) and binding sites between LPA and different LPARs in bovines.

Molecules	Total Score	Crash Score	Polar Score	Binding Sites
LPAR1	9.9848	−6.4532	1.8122	GLN-125, ASP-129
LPAR2	11.4654	−2.3193	4.2882	TYR-85, GLN-108, ASP-112, TRP-193, GLN-258
LPAR3	7.1863	−6.5855	2.1880	LYS-275
LPAR4	9.4880	−1.1479	2.9762	CYS-111, GLY-115, CYS-188, GLU-190
LPAR5	8.6331	−3.3448	2.7064	ARG-77, GLN-160
LPAR6	6.8475	−3.0658	3.9243	GLU-167, LYS-175
GPR87	13.2981	−5.6552	5.9959	ARG-97, LYS-110, SER-118, CYS-192, TYR-293, LYS-296
P2RY10	10.0559	−2.2346	4.5339	TYR-25, TYR-82, TYR-102, HIS-276

**Table 2 ijms-26-02596-t002:** List of primers used for quantitative RT-PCR.

Gene Name	Accession	Direction	Sequence	Annealing Temperature (°C)	Size (bp)
*LPAR1*	NM_174047.2	Forward Reverse	GTTCAACACAGGGCCCAATACT AACCGTCAGGCTGGTGTCA	60	85
*LPAR2*	NM_001192235.1	Forward Reverse	ATCGCCGCTTCCACCAACCCAT GCCGTCAGGCTCGTGTCCAACA	63.1	172
*LPAR3*	NC_007305.3	Forward Reverse	TCAAGTAAAGCACGGGCAGAA CGGGTGAGAACGCATTGTG	60	90
*LPAR4*	NM_001098105.1	Forward Reverse	ATCACCAATCTGGCCCTCTCT CAGTGGCGGTTGAAATTGTAAAA	60	80
*LPAR5*	NM_001034304.2	Forward Reverse	AGAAGCCAGCTCGGGAGA CCAGAGCAAACGGAGGTTCA	60	161
*LPAR6*	NM_001101284.1	Forward Reverse	TGCTCAGCAGCAATAGCAGT CCCAGTAAAGTCCTGGCGTT	60	284
*GPR87*	NM_001205451.1	Forward Reverse	CCAGCAGGCAATTCATAAGCCAGTC TCGGATGCTTTCGCTCCTCGTTC	58.9	311
*P2RY10*	NM_001075699.2	Forward Reverse	ACCAACTCCCTGCAAAGAAAC GTGGTTGCGTAGAGAGGGTAA	60	171
*BAX*	NM_173894.1	Forward Reverse	GCAGAGGATGATCGCAGCTG CCAATGTCCAGCCCATGATG	62	197
*BCL2*	XM_005224105.5	Forward Reverse	CTTCGCCGAGATGTCCAGTC CACCACCGTGGCGAAGC	58	70
*ATX*	NM_001080293.1	Forward Reverse	ACCCCCTGATTGTCGATGTG TCTCCGCATCTGTCCTTGGT	60	120
*PLA2*	NM_001075864.1	Forward Reverse	CCATTATTCCCACAAGTTCACAGTA TGTCAAGCATGTCACCAAAGGT	60	80
*YWHAZ*	XM_025001430.1	Forward Reverse	GCATCCCACAGACTATTTCC GCAAAGACAATGACAGACCA	58	120

## Data Availability

Data are contained within the article.

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
