# Peer review of "Evaluation of Lysophosphatidic Acid Effects and Its Receptors During Bovine Embryo Development"

_ijms, 2025, doi:10.3390/ijms26062596_

Round 1
Reviewer 1 Report
Comments and Suggestions for Authors
The study is an extension of the investigators' previous work that examine the impact of LPA on bovine embryonic development. In the present study, the investigators assessed the presence of LPA in reproductive tract/cells and confirm the presence of LPA in the oviductal fluid. They also tested the impact of LPA supplementation on embryo quality (cell number and apoptosis) as well as expression of LPA receptors at different stage of embryonic development and their binding affinity. Overall, the experiment is logical and well executed. The manuscript is clearly written. The findings provide fundamental insights into the roles and mechanisms of LPA in embryonic development.
Author Response
Comments 1: The study is an extension of the investigators' previous work that examine the impact of LPA on bovine embryonic development. In the present study, the investigators assessed the presence of LPA in reproductive tract/cells and confirm the presence of LPA in the oviductal fluid. They also tested the impact of LPA supplementation on embryo quality (cell number and apoptosis) as well as expression of LPA receptors at different stage of embryonic development and their binding affinity. Overall, the experiment is logical and well executed. The manuscript is clearly written. The findings provide fundamental insights into the roles and mechanisms of LPA in embryonic development.
Response 1: We are deeply grateful to the reviewer for thoughtful evaluation of our work and the recognition of the scientific rigor in experimental design. To improve readability, we have implemented comprehensive linguistic improvements throughout the manuscript and the changes and corrections are highlighted in the revised manuscript.
In the revised manuscript, we have substantially expanded the discussion section (now increased from 667 to 944 words) and all the results are explicitly discussed. Future work could focus on using siRNA to knock down LPAR2 and studying LPAR2 function. In addition, we added binding sites information between LPA and LPAR in Table 1. It would be interesting to mutate one or more codons encoding the amino acid of the binding sites, providing us clues on how LPA interacts with LPAR2. We added this information to the revised manuscript (line 266-270). Moreover, it is of great importance to evaluate bovine embryo quality after LPA treatment through embryo transfer and live birth validation. This is a critical step for bridging our in vitro findings to clinical applications in livestock reproduction (line 225-228).
Thank you again for your encouraging comments!

Reviewer 2 Report
Comments and Suggestions for Authors
- Some grammars and writings should be revised and polished. Here are some examples in the abstract
- line 21, …to assess the presence….….
Ex. line 22-23, …. and determine cell apoptosis in embryos after LPA stimulation by TUNEL assay and quantitative RT-PCR.
Ex. line 25, An increase of total cell number…....
Ex. line 27-28, in embryos after the 16-cell stage in RNA-seq and quantitative RT-PCR analysis.
Ex. line 28-29, only the expression of LPAR2 was significantly increased…..
Ex. line 31, …enrich information of related signaling mediators during embryonic development.
- In Figure 1, the authors collected ovaries from a slaughter house and measured LPA in oviductal fluid, follicular fluid, uterine fluid and serum and expression of ATX and PLA2 in OEC, GC, and UEC. Why not further separated follicular fluid and GC from different follicle classes and serum into different estrus stages to make the LPA level fluctuations with more physiological significance?
- In Figure 2, where are the results of the concentration of LPA in oviductal fluid, follicular fluid, uterine fluid and serum?
- In Table 1, the species of LPAR for their binding scores with LPA should be noted, bovine? human or mouse?
- In Figure 4, the authors stated that results are derived from RNA-seq data in reference 24 and 25. The authors need to describe how the data and result curation from the dataset of the references. This critical to avoid repetitive publication of the same data.
The English could be improved to more clearly express the research.
Author Response
Comments 1: Some grammars and writings should be revised and polished.
Response 1: We sincerely appreciate the reviewer's careful reading and valuable suggestions regarding language refinement. In response to this constructive feedback, we have implemented comprehensive linguistic improvements throughout the manuscript and the changes and corrections are highlighted in yellow in the revised manuscript.
Comments 2: In Figure 1, the authors collected ovaries from a slaughterhouse and measured LPA in oviductal fluid, follicular fluid, uterine fluid and serum and expression of ATX and PLA2 in OEC, GC, and UEC. Why not further separated follicular fluid and GC from different follicle classes and serum into different estrus stages to make the LPA level fluctuations with more physiological significance?
Response 2: We are grateful for the reviewer's insightful suggestion. We fully agree that it would be much more physiological significance if we measured LPA in different follicle classes or different estrus stages. In our manuscript, we intended to evaluate LPA effects during bovine embryonic development and focused on LPA concentration in oviductal fluid, which is considered as the most suitable microenvironment for early embryo development. This data provides a clue that low quality of IVP embryos might be related to the absence of LPA during embryo culture. LPA concentration in follicular fluid, uterine fluid and serum were set as control.
The investigation of LPA concentration variations across follicular size categories indeed presents intriguing scientific merit, particularly in the context of oocyte maturation. However, this aspect falls a bit beyond the scope of our current research emphasis on LPA effects during embryonic development. As for serum into different estrus stages, we feel so sorry that cow ovaries are mixed before they arrived and we can not get information of estrus stage before cow are slaughtered, based on regulations of the slaughterhouse.
We still think it is important to add this valuable information to the manuscript. We have added one paragraph to discuss the presence of LPA in oviductal fluid and other fluids in Line 203-212.
Comments 3: In Figure 2, where are the results of the concentration of LPA in oviductal fluid, follicular fluid, uterine fluid and serum?
Response 3: Thank you for pointing this out and we feel sorry for the mistake we made. The results of the concentration of LPA in oviductal fluid, follicular fluid, uterine fluid and serum have been shown in Figure 1. We deleted this sentence in the legend of Figure 2.
Comments 4: In Table 1, the species of LPAR for their binding scores with LPA should be noted, bovine? human or mouse?
Response 4: Thank you for pointing this out and it is indeed very important to state the species of LPAR. Therefore, we add specie information in the figure legend (Line 151).
Comments 5: In Figure 4, the authors stated that results are derived from RNA-seq data in reference 24 and 25. The authors need to describe how the data and result curation from the dataset of the references. This critical to avoid repetitive publication of the same data.
Response 5: We appreciate the reviewer's astute observation regarding the proper citation and data processing. To distinguish from the original data set, we presented our reanalyzed data in FPKM, not the repetitive publication of the same data. The RNA-seq analyses presented in Figure 4 were conducted through the following rigorous process: Raw RNA‐seq data were obtained from GEO Series GSE52415 (in vitro embryos) and GSE59186 (in vivo embryos). Quality of raw sequence data was checked using FastQC (v1.0.0), followed by low-quality reads and adapter sequences removal using FastP (v0.24.0). The filtered clean reads were aligned to the Bos taurus reference genome ARS-UCD1.3 (GCA_002263795.3) using Hisat2 (v2.1.0). Subsequently, the resulting SAM files were read and converted to BAM files using Samtools (v1.21). The read counts matrix was calculated using featureCounts (v 2.0.5). Fragments per kilobase of transcript per million mapped reads (FPKM) values of genes were calculated in edgeR (version 4.4.2). We added this information to the revised manuscript (line 348-356).

Reviewer 3 Report
Comments and Suggestions for Authors
The work is interesting, and the experiments are well designed according to the stated objectives. However, there are some aspects of the writing that need improvement, which I will list below:
line 19: "in species", in which ones? or add in several or many species...
line 58: "specific G-protein couple receptor" is repeated
line 74: rewrite sentences, it is not clear
line 86: change molecule for "molecular"
lines 138: change "have not studied in bovine" for have not been studied...
improve figure 2 references: it is not clear what the 2 different column colors represents. specially for figure 2 B to F.
In addition, the discussion is brief, and not all observed results are analyzed. For example, there is no discussion on why differences are observed in the expression profiles of the various LPA receptors between embryos produced in vitro and those obtained in vivo. Another result that was not discussed, is the expression patter of the enzymes that produce LPA in the different tissues analyzed.
A suggestion that would add value to the work would be, to confirm the role of LPAR2, identified as the main effector of LPA action, by analyzing how would the embryo development if its action were inhibited, either by using an inhibitor or a specific antagonist for this receptor.
On the other hand, since it is suggested that the use of LPA improves the quality and viability of bovine embryos, an ultimate goal would be to confirm this by transferring these embryos to recipient females and verifying a higher pregnancy rate and the birth of live offspring
Comments on the Quality of English Languagethe work is well written overall, however there some phrases o parts of some paragraph that need to be improve as i mention to the authors.
Author Response
Comments 1: The work is interesting, and the experiments are well designed according to the stated objectives. However, there are some aspects of the writing that need improvement, which I will list below.
Response 1: We sincerely appreciate the reviewer's careful reading and valuable suggestions regarding language refinement. In response to this constructive feedback, we have implemented comprehensive linguistic improvements throughout the manuscript and the changes and corrections are highlighted in yellow in the revised manuscript.
Comments 2: improve figure 2 references: it is not clear what the 2 different column colors represents. specially for figure 2 B to F.
Response 2: We sincerely appreciate this valuable feedback regarding figure clarity. In response to the reviewer's comment, we have changed the legend of Figure 2 to illustrate column color representation in the revised manuscript (line126-127).
Comments 3: In addition, the discussion is brief, and not all observed results are analyzed. For example, there is no discussion on why differences are observed in the expression profiles of the various LPA receptors between embryos produced in vitro and those obtained in vivo. Another result that was not discussed, is the expression patter of the enzymes that produce LPA in the different tissues analyzed.
Response 3: We are grateful for the reviewer's insightful critique and not all the results were discussed. In the revised manuscript, we have substantially expanded the discussion section (now increased from 667 to 944 words) with the following targeted enhancements: 1. different expression profiles of the various LPA receptors between in vitro and vivo embryos (line 242-246). 2. Expression pattern of the LPA producing enzyme (line 203-212).
Comments 4: A suggestion that would add value to the work would be, to confirm the role of LPAR2, identified as the main effector of LPA action, by analyzing how would the embryo development if its action were inhibited, either by using an inhibitor or a specific antagonist for this receptor.
Response 4: We sincerely thank you for raising this critical point regarding the functional validation of LPAR2 through inhibition experiments. We fully agree that such experiments would strengthen the mechanistic conclusions of our study. There is no commercial LPAR2 inhibitor or antagonist for bovine and the effect of inhibitor needs to be verified first. Another approach is to knock down expression of LPAR2 using siRNA. In addition, we added binding sites information between LPA and LPAR2 in Table 1. It would be interesting to mutate one or more codons encoding the amino acid of the binding sites, providing us clues on how LPA interacts with LPAR2. We added this information to the revised manuscript (line 266-270).
Comments 5: On the other hand, since it is suggested that the use of LPA improves the quality and viability of bovine embryos, an ultimate goal would be to confirm this by transferring these embryos to recipient females and verifying a higher pregnancy rate and the birth of live offspring.
Response 5: We sincerely appreciate the reviewer’s insightful suggestion to evaluate bovine embryo quality after LPA treatment through embryo transfer and live birth validation. This is indeed a critical step for bridging our in vitro findings to clinical applications in livestock reproduction. However, due to limited funding and project timeline, it is not possible for us to perform embryo transfer experiments. It is indeed one of the limitations in this study. Still, we explicitly discussed the importance of embryo transfer experiments for embryo quality validation in the revised manuscript (line 225-228).

Round 2
Reviewer 2 Report
Comments and Suggestions for Authors
The responses are accepted
Author Response
We sincerely appreciate the reviewer's careful reading and valuable suggestions. We have conducted a thorough review of the manuscript and identified additional areas requiring editorial refinement. All modifications and corrections have been clearly highlighted in yellow within the revised document.
Revised Section 2.5. Expression of LPARs in In Vitro Embryo after LPA Treatment (from line 181):
Expressions of LPAR1, LPAR2, LPAR3 and GPR87 were detectable (Figure 5A-D), but LPAR4 and LPAR5 transcripts remained undetectable throughout the examined developmental stages, which were consistent with the RNA-seq results. Notably, we identified a discrepancy between two detection methods: while LPAR6 expression was consistently observed in both in vivo and in vitro derived embryos in RNA-seq data, P2RY10 mRNA was exclusively detectable as determined by qRT-PCR (Figure 5E). A particularly significant finding emerged in LPAR2 expression analysis. The expression of LPAR2 was significantly higher in day 6 LPA stimulated blastocysts compared to control blastocysts and its expression increased significantly from day 5 morulae to day 6 blastocysts after LPA treatment (Figure 5B).
Revised Line 223:
Original: "...more total-cell number..."
Revised: "...high embryonic cell count..."
Revised Line 239:
Original: "with GPR87, LPAR2 and P2Y10 emerging as top receptor candidates of LPA with total score more than 10."
Revised: "with GPR87, LPAR2 and P2RY10 exhibiting the highest interaction potential (total score >10)."
Revised Section 4.6. Molecular docking (from line 343):
The crystal structures of the target proteins were subsequently simulated using the Swiss-model. The target proteins were then refined using Sybyl-X 2.0 software (Tripos, St. Louis, Missouri, USA), which involved the removal of water molecules and metal ions, the addition of hydrogen atoms, the correction of any missing amino acid residues within the crystal structure and other structural optimizations. LPA structure was obtained from the PubChem database. Evaluation of the direct binding capacity of LPARs to LPA was performed using Sybyl-X 2.0. Visualization of molecular docking results were carried out using PYMOL website (www.pymol.org).
Reviewer 3 Report
Comments and Suggestions for Authors
the edits made are correct, as well as the responses to the concerns raised. however, there are some more edits and revisions that should be made before published, which I detail below:
- in section 2.5. Expression of LPARs in In Vitro Embryo after LPA treatment, rewrite the paragraph from line 181, as it is not clearly understood.
- line 223 "...more total-cell number..." it is not well expressed
- line 237 "score more than 10" it is not well expressed
The work may be suitable for publication; however, some paragraphs or sections are still difficult to understand that need to be imptoved.
Author Response
Comments 1: the edits made are correct, as well as the responses to the concerns raised. however, there are some more edits and revisions that should be made before published, which I detail below:
Response 1: We sincerely appreciate the reviewer's careful reading and valuable suggestions regarding language refinement. We have implemented comprehensive linguistic enhancements across the entire manuscript. All modifications and corrections, particularly those in the sections you specifically mentioned, have been clearly highlighted in yellow within the revised document.
Revised Section 2.5. Expression of LPARs in In Vitro Embryo after LPA Treatment (from line 181):
Expressions of LPAR1, LPAR2, LPAR3 and GPR87 were detectable (Figure 5A-D), but LPAR4 and LPAR5 transcripts remained undetectable throughout the examined developmental stages, which were consistent with the RNA-seq results. Notably, we identified a discrepancy between two detection methods: while LPAR6 expression was consistently observed in both in vivo and in vitro derived embryos in RNA-seq data, P2RY10 mRNA was exclusively detectable as determined by qRT-PCR (Figure 5E). A particularly significant finding emerged in LPAR2 expression analysis. The expression of LPAR2 was significantly higher in day 6 LPA stimulated blastocysts compared to control blastocysts and its expression increased significantly from day 5 morulae to day 6 blastocysts after LPA treatment (Figure 5B).
Revised Line 223:
Original: "...more total-cell number..."
Revised: "...high embryonic cell count..."
Revised Line 239:
Original: "with GPR87, LPAR2 and P2Y10 emerging as top receptor candidates of LPA with total score more than 10."
Revised: "with GPR87, LPAR2 and P2RY10 exhibiting the highest interaction potential (total score >10)"
Revised Section 4.6. Molecular docking (from line 343):
The crystal structures of the target proteins were subsequently simulated using the Swiss-model. The target proteins were then refined using Sybyl-X 2.0 software (Tripos, St. Louis, Missouri, USA), which involved the removal of water molecules and metal ions, the addition of hydrogen atoms, the correction of any missing amino acid residues within the crystal structure and other structural optimizations. LPA structure was obtained from the PubChem database. Evaluation of the direct binding capacity of LPARs to LPA was performed using Sybyl-X 2.0. Visualization of molecular docking results were carried out using PYMOL website (www.pymol.org).